# Impact of Combined Thermal Pressure Treatments on Physical Properties and Stability of Whey Protein Gel Emulsions

**DOI:** 10.3390/foods12132447

**Published:** 2023-06-22

**Authors:** Marcello Alinovi, Massimiliano Rinaldi, Maria Paciulli, Francesca Bot, Davide Barbanti, Emma Chiavaro

**Affiliations:** Department of Food and Drug, University of Parma, Parco Area delle Scienze Pad. 33 (Tecnopolo), 43124 Parma, Italy; massimiliano.rinaldi@unipr.it (M.R.); maria.paciulli@unipr.it (M.P.); francesca.bot@unipr.it (F.B.); davide.barbanti@unipr.it (D.B.)

**Keywords:** emulsion gels, whey proteins, physical stability, particle size, rheology, fat blocker, fat replacer, homogenization

## Abstract

Emulsion gels are gaining interest as fat replacers due to their benefits associated with calorie reduction and their versatility in a wide range of products. Their production process needs to be tailored to obtain the desired stability and physicochemical properties. This study investigated the effect of heat (70, 80, and 90 °C) and pressure (5, 10, and 15 MPa) to produce whey protein emulsion gels using a pilot-scale tubular heat exchanger equipped with a homogenization valve. Both temperature and pressure determined a significant effect (*p* < 0.05) on the rheological moduli, with the treated samples displaying a predominant elastic behavior. The treatments also showed an improved pseudoplasticity due to the significant reduction in the flow behavior index (*p* < 0.05). All the samples showed a bimodal particle size distribution; by increasing the temperature up to 80 °C, a reduction in Dv50 (50th percentile) values compared to the control samples was observed. At 90 °C, the Dv50 value increased because of coalescence and flocculation phenomena occurring during or immediately after processing. The greater aggregation and structural development obtained with stronger process conditions improved the stability of the emulsions. The results show the capability to produce gel emulsions with good physical properties that could be proposed as food ingredients to substitute fats in food products.

## 1. Introduction

Cardiovascular diseases caused by a high intake of trans fats are one of the worldwide causes of morbidity and mortality [1]. In particular, it has been demonstrated that a high consumption of foods containing saturated or trans fats increases the prevalence of chronic diseases, such as obesity, hypertension, and coronary heart disease [2,3]. To tackle these challenges, food products with a reduced content of saturated solid fats have been developed using different technological approaches including oleo gels and gelled emulsions [4,5,6].

Emulsion gels, soft complex colloidal materials formed by matrices of polymeric gels into which oil droplets are incorporated, are made through the gelation of a polymer (i.e., proteins) in the continuous phase of an emulsion with a low or moderate oil content (generally lower than 30%) [7,8,9]. These food systems have technological properties suitable in a wide range of applications, such as cream cheese, yoghurt, and bakery products. In addition, protein-based emulsion gels can be used to modify flavor release [3], to protect active ingredients [10,11], and to increase nutraceutical bioavailability [8]. 

The methods generally used to prepare protein-based emulsion gels are based on treatments causing the aggregation of free proteins in the continuous phase, the aggregation of protein-coated oil droplets, or a combination of both [7]. The most common approaches to develop protein-based emulsion gels in the food industry are heat treatment, pH adjustment, mineral ion addition, biopolymer-induced flocculation, and enzyme treatment [12,13,14,15]. More specifically, in systems containing whey proteins, gelation can be induced by heating the emulsion above the denaturation temperature, which causes the proteins to unfold and aggregate [16]. The final structures of whey protein-stabilized emulsions can be tuned from liquid-like to solid-like states by changing the conditions of the pH, ionic strength, and heat treatment, i.e., factors that control the degree of protein unfolding and aggregation [7]. Several other factors affect the denaturation–aggregation process, such as protein concentration, and by generally increasing the concentration, a high reaction rate and bigger aggregate can be obtained [17,18]. Heating time and temperature are important process parameters influencing the denaturation–aggregation process with a direct effect on the denaturation–aggregation degree. In addition, a decrease in aggregate diameter with an increasing shear rate during continuous processes was also reported due to the reduced efficiency of the collisions at a high shear rate [18]. When a continuous thermal treatment of WP is performed, the induced protein gelation affects flow properties and heat transfer [19]. These processes are coupled; on one hand, the product transformation depends on the operation conditions (velocity, shear rate, and temperature). Knowledge of WP behavior as a function of variable shear rate is also crucial to industrialize the production of some protein-based dairy products, such as ricotta, favoring the development of the continuous operation mode [20] without negatively affecting the product rheology. Due to the complexity of the above-mentioned phenomena, there is an urgent need for a better understanding of whey protein denaturation phenomena, together with process design and optimization in order to reduce processing time and cost [21].

Generally, emulsion-filled gels are prepared by premixing aqueous solutions with high-speed mixers and after prehomogenization by passing them through a high-pressure homogenizer. However, high-pressure treatments could represent a limit for the industrial production of these products, while conventional homogenization processes using moderate pressure (up to 20 MPa) within the dairy sector have shown a huge potential in stabilizing emulsions [22].

In this context, the aim of this work was to evaluate the influence of homogenization and heat treatment on the formation and physical stability of WP emulsion gels. To this end, the effect of different WPI denaturation degrees on the O/W emulsion stability was evaluated by varying the heat treatment temperature (70, 80, and 90 °C) and the homogenization pressure (5, 10, and 15 MPa). The results of this work will enhance our understanding on the potential of whey protein emulsion gels in formulating and developing food products with a lower content of fat and tailored physical properties and stability.

## 2. Materials and Methods

### 2.1. Materials

Whey protein concentrate (WPC) (protein 80% *w*/*w*, water 5% *w*/*w*, lactose 7% *w*/*w*, and fat 4.6% *w*/*w* according to product specification) was kindly donated by Molkerei Meggle Wasserburg GmbH & Co.KG (Wasserburg, Germany). High-oleic sunflower vegetable oil was obtained from a local market (Conad, Parma, Italy).

### 2.2. Emulsion and Gel Preparation

#### 2.2.1. Stock Solutions

WPC stock solutions (20% *w*/*w*) were prepared by powder dissolution in deionized water under mild magnetic stirring (150 rpm) for 2 h at room temperature; subsequently, the stock solution was kept at 4 °C for 24 h without stirring. The day after the reconstitution step, WP solution was de-aerated by means of a vacuum oven (ISCO, Milan, Italy) at constant temperature of 25 °C for 30 min.

#### 2.2.2. Preparation and Processing of WP Emulsions

A coarse emulsion was prepared by mixing 20% (*w*/*w*) sunflower oil with the stock protein solutions (see Section 2.2.1) with a high-speed mixer (Sammic, Azkoitia, Gipuzkoa, Spain) at 9000 rpm for 30 s to reach a final WP concentration of 8% (*w*/*w*). The coarse emulsion (control) was processed by using a lab-scale pilot plant (Valfor Srl, Parma, Italy) by three different treatment temperatures (70, 80, and 90 °C for 138 s) and operating pressures (5, 10, and 15 MPa). Briefly, the pilot plant was assembled with a feeding tank, a volumetric piston pump (OBL Srl, Milan, Italy) set at a flow rate of 20 L/h, and a heat exchange section. This section was made of 3 spiral coils of stainless steel pipe (nominal diameter = 6 mm, wall thickness = 1 mm, and length = 27.03 m) submerged into two water-stirred thermostatic baths (Julabo Italia Srl., Milan, Italy). The total internal volume of the spiral coil heating zone was 0.76 L. Further details about the pilot heat exchanger design can be found in Rinaldi et al. [19]. At the outlet of the heating section, a lamination valve (Valfor Srl, Parma, Italy) was mounted in order to exert the above-mentioned pressures on the samples.

### 2.3. Analytical Determinations

#### 2.3.1. Rheological Measurements

Rheological measurements of the emulsions were performed using a stress-controlled rheometer (mod. MCR 302, Anton Paar, Graz, Austria) equipped with a plate/plate system (50-mm diameter and 1.000 mm gap) and a Peltier convection system (mod. CTD 180, Anton Paar, Graz, Austria) to control the temperature of the samples (25 °C). Frequency sweep experiments were conducted using a frequency ranging from 0.1 to 100 s^−1^ and a 0.15% strain that fell within the linear viscoelastic region (LVR) of each sample, previously determined by performing strain sweep experiments. Flow rate tests were carried out for the determination of the flow behavior and of the hysteresis area by using the hysteresis loop technique, where the shear rate is increased linearly from 0.1 to 100 s^−1^, maintained for 30 s, and then decreased to 0.1 s^−1^ at the same rate during a single isothermal rheological experiment. Data obtained from cycle were fitted by means of the Ostwald–de Waele power law model (Equation (1)):σ = k∙γ^n^
(1)
where k is the consistency coefficient (Pa s^n^) and n is the flow behavior index (dimensionless). Both the rotational and oscillatory analyses were performed in triplicate.

#### 2.3.2. Particle Size Distribution

Particle size distribution (PSD) of the coarse and emulsion gels was measured using a laser light diffraction unit (Spray Tech, Malvern Instruments Ltd., Worcestershire, UK). Emulsion gels were introduced into the dispersing unit of the instrument using ultrapure water as dispersant until a laser obscuration of 12 ± 1% was achieved. Analysis of PSD was performed using the spherical model, with particle refractive index of 1.46 for the dispersed phase (sunflower oil) and absorption of 0.001. The refractive index for the dispersant (water) was set to 1.33. The distributions were also interpreted by considering the volume-weighted distribution percentiles (Dv10, Dv50, and Dv90) and the volume mean particle diameter (D_[3,4]_). The particle size analyses were performed at least in duplicate.

#### 2.3.3. Physical Stability 

The creaming stability was measured by centrifuging at 6500 rpm for 30 min at room temperature using a benchtop centrifuge (mod. 5810 R, Eppendorf, Hamburg, Germany). Quantification of separated water (expressed as% *w*/*w*) was determined as described by Surh et al. [23] and Mutilangi et al. [24]. Emulsion stability was also tested by means of four repeated freezing–thawing cycles, as described by Aoki et al. [25]. Briefly, samples (10 mL) were incubated at −20 °C for 22 h and subsequently thawed in a water bath at 30 °C for 2 h. The effect of repeated freezing–thawing cycles on the emulsion stability was quantified by measuring the percentage of separated oil just after each cycle. Both sedimentation/centrifugation and freezing–thawing tests were performed in duplicate.

### 2.4. Statistical Analysis 

For all the collected analytical data, a two-way analysis of variance (ANOVA) was performed by means of SPSS statistical software (Version 27.0, IBM., Chicago, IL, USA). Temperature and pressure were considered as main effects of the model; temperature x pressure was included as the interaction term of the model. Significant main effects and interaction were declared at a *p* < 0.05. A Tukey HSD post hoc test (α = 0.05) was also performed for pairwise multiple comparisons among different groups, also including the control sample in the comparison.

## 3. Results and Discussion

### 3.1. Rheological Characteristics of WP Emulsion Gels

Dynamic oscillatory and rotational rheological characteristics are important quality attributes of fat replacers, as their functionality as ingredients in food formulations, such as baked products, is primarily due to their mechanical properties [26]. The results of the rotational analysis (Ostwald–de Waele coefficients *n* and *k*, hysteresis area) are reported in Table 1. The control (coarse emulsion) showed a very different flow behavior compared to the emulsion gels produced at increasing temperatures and pressures. In particular, the coarse emulsion had the lowest and the highest consistencies (*k* ~0.29 mPa·s) and flow behavior (*n* ~0.74) indexes compared to the other samples. These results were similar to those reported by Hebishy et al. [27], who analyzed a WP-stabilized vegetable oil emulsion (50% oil fraction) made with conventional homogenization (15 MPa). Both pressure and temperature showed a significant effect (*p* < 0.05) on the different rheological parameters. By imposing the selected temperature conditions (70, 80, and 90 °C) in the heating section of the pilot plant, the flow behavior of the samples changed considerably. The thermally treated samples showed an improved pseudoplastic behavior due to the significant reduction (*p* < 0.05) in the flow behavior index *n* (*p* < 0.05) with values between 0.490 and 0.292, and a significant increase in the consistency index (*p* < 0.05) up to 255 mPa∙s in the sample treated at 90 °C and 15 MPa. These rheological modifications can be attributed to the heat-induced formation of WP aggregates and of a protein intermolecular network that increases the flow resistance of the emulsion [28] especially at low shear rates; on the contrary, at high shear rates, the polymeric chains of the aggregates become more and more coordinated to the flux and offer a relatively lower flow resistance. The hysteresis area of the samples showed a thixotropic behavior, with a higher hysteresis area in the thermally treated samples compared to the control. This result can be directly related to the amount of mechanical energy used to shear-induce the breakdown of the protein network, as reported by other authors [29,30]. The results also show that the homogenization step after the thermal treatment induced a significant (*p* < 0.05) change in the flow behavior of the emulsions for some of the analyzed samples; in particular, for the samples treated at 90 °C, the increase in the homogenization pressure improved the consistency index *k* (~ +407% from 5 to 15 MPa) and the hysteresis area (~ +645% from 5 to 15 MPa), while for the samples treated at 70 and 80 °C, this was less pronounced and/or not significant (*p* > 0.05). The results indicate a significant (*p* < 0.05) interaction between temperature and pressure, meaning that the homogenization step promotes the formation of different rheological structures when coupled to different temperature conditions [30]. It is generally accepted that WP unfolds within a temperature range between 60 and 90 °C, causing the exposure of hydrophobic regions that can interact with oil droplets and form viscoelastic layers at the interface [31]; after protein unfolding, the formation of WP aggregates or protein structural networks is immediate, but it is time- and temperature-dependent [30,32]. Sliwinski et al. [32], studying the gelation properties of WP emulsions, observed a peak in the viscosity at 90 °C after 6–8 min of heating time, while for WP emulsions treated at 75 °C, the viscosity increase reached a plateau after longer heating times (~40 min). It may be hypothesized that the application of different homogenization pressures on previously thermally unfolded WP (unfolded at different ratios, in relation to the different heating processes) can have an impact on the protein–protein and protein–oil interface interactions, thus modifying the rheological characteristics of the emulsion gels. In addition, a greater reduction in the oil droplets’ diameter at higher pressure conditions can affect the rheological changes of the system, as WP-covered oil droplets can be classified as active filler particles that are able to improve the gel strength [33].

To have a better understanding of the rheological properties of the WP emulsion gels, their dynamic rheological characteristics (Figure 1A–C) were measured. The control sample showed, as expected, a predominant viscous behavior behaving as a liquid-like emulsion, as the loss modulus G″ was always higher than the elastic modulus G′ (tanδ > 1) (Figure 1C). On the other hand, WP emulsion gels showed a predominant elastic behavior, with G′ > G″ and tanδ < 1, indicating the formation of a structured and cross-linked gelled system. Both temperature and pressure had a significant effect (*p* < 0.05) on the rheological moduli G′ and G″. In particular, both moduli showed an increase at higher treatment temperatures with a more pronounced increase from 70 °C to 80°C (~+9000% for G′) than from 80 °C to 90 °C (~ +107% for G′). This indicates a significant increase (*p* < 0.05) in the WP aggregation rate, as above 70 °C the denaturation of the adsorbed WP may lead to enhanced interdroplet interactions thus increasing the gel strength and the solid-like behavior of the system together with the reduction in tanδ [34]. The increase in the applied homogenization pressure generally caused a significant increase (*p* < 0.05) in both the rheological moduli. In accordance with what was observed from the rotational rheological analyses, the effect of the homogenization pressure was significantly influenced (*p* < 0.05) by the temperature during the heating phase (significant temperature–pressure interaction, *p* < 0.05). 

### 3.2. Particle Size Measurements of WP Emulsion Gels

Detailed information on the effect of temperature and pressure on WP emulsion gels can be obtained by measuring the particle size distribution (PSD) (Figure 2 and Table 2). The control samples showed a bimodal volume-based PSD with a Dv50 of 10.10 μm and Dv10 and Dv90 of 1.58 and 41.14 μm, respectively (Table 2). Furthermore, in the case of emulsion gels, a bimodal PSD can be observed in all the samples (Figure 2); the presence of a bimodal PSD is probably caused by the occurrence of an oil emulsion dispersed in a sheared gel, consisting of microgels and gelled particles of WP with bigger dimensions [35]. Emulsion gels treated at temperatures up to 80 °C showed a significantly lower (*p* < 0.05) Dv50 value than the control samples. In the samples treated at 70 and 80 °C, the Dv50 ranged between 8.82 and 2.56 μm (Table 2). In addition, no significant (*p* > 0.05) differences in the Dv10 and Dv90 values can be observed between the control and treated samples. Overall, the results indicate that by increasing the temperature to 70 °C and setting a low-pressure treatment (i.e., 5 and 10 MPa), it is possible to obtain emulsion gels with lower Dv50 values than the coarse emulsion. However, an increment in the pressure (i.e., 15 MPa) caused the coalescence/aggregation phenomena with an increase in the Dv50 values (i.e., 8.82 μm). In addition, by increasing the temperature up to 90 °C, a significant (*p* < 0.05) increase in the Dv50 values was observed at stronger pressure treatments, with values ranging between 11.79 and 22.66 μm (Table 2). The increase in the particle size distribution and related parameters can be attributed to coalescence and flocculation phenomena mediated by interdroplet interactions, which are caused by the unfolding of WP adsorbed on the droplet interface [34] together with droplet–droplet, protein–protein, and droplet–protein aggregation mechanisms [16]. Similar results have been observed in whey protein isolate-based emulsions (20% oil) with increasing total solids [16,36].

### 3.3. Physical Stability of WP Emulsion Gels

Sedimentation (% of water separated by centrifugation) and creaming stability (% of oil separated by repeated freezing–thawing cycles) are reported in Table 3. The control (coarse emulsion) showed a very low stability both when a centrifugal force was applied (% of separated water: 97.85 ± 0.21%) and when the sample was subjected to freezing–thawing cycles (% of separated oil: 97.28 ± 0.40%), due to the absence of both the mild-pressure homogenization phase and the heat treatment. Concerning the treated samples, it was possible to highlight significant main effects and interactions of temperature and pressure (*p* < 0.05) on both the sedimentation and creaming stability results. In particular, the physical stability of the emulsion improved both as a function of the applied temperature and pressure in the pilot plant. In the samples treated at 70 °C, the sedimentation stability ranged from ~26.8% to ~11.9%, depending on the pressure applied at the end of the heating phase (from 5 to 15 MPa, respectively). In addition, the creaming stability of the above-mentioned emulsion gels increased at higher pressure treatments, as the amount of separated oil ranged from ~73.8% at 5 MPa to 59.4% at 15 MPa. The exerted pressure may also have an impact on the structural organization of the WP aggregates that form during the continuous heat treatment, thus inducing differences in the physical stability of the final systems [37].

Higher temperature treatments (80 and 90 °C) before the homogenization step caused a strong improvement in the sedimentation stability, as the amount of separated water following centrifugation was lower than 2.5%; in particular, the amount of separated water reached values close to 0% for the treatments at 15 MPa (~0.6% for the sample treated at 80 °C, and ~0.1% for the sample treated at 90 °C). The greater sedimentation stability observed at higher temperatures, which highlights an improved water-holding capacity of the system, can be related to the formation of disulphide bonds within WP molecules (mainly β-lactoglobulin, but also α-lactalbumin and bovine serum albumin) that cause the formation of protein aggregates [38,39]. The formation of intermolecular interactions causes a rheological improvement in the gel network and a greater capacity of the gel structure to retain water molecules. Accordingly, Boye et al. [40], who studied the effects of different factors (i.e., heating time, temperature, pH, NaCl, and sucrose) on the gelling behavior of WPC solutions, observed a sharp increase in gel strength and water-holding capacity when the heating temperature was above 70 °C. Hebishy et al. [41] evaluated the effect of high-pressure homogenization (i.e., 100 and 200 MPa (single-stage)) on the stability of WP emulsions and observed a high creaming stability (~17 days) as a possible consequence of the reduction in the particle size of the oil droplets.

Similarly, as reported in Table 3, there was an increase in the creaming stability when the samples were treated at 80 °C, reaching 27.5 g of oil/100 mL of sample. This parameter was also strongly influenced by the pressure applied during the homogenization step: in the case of the sample obtained at 70 °C and 90 °C, a significant reduction (*p* < 0.05) in separated fat was detected. This result can be attributed to a stronger interaction of the oil interface with WP. Zamora et al. [42], analyzing milk homogenized at 18 MPa at 60 °C and pasteurized at 72 °C for 15 s, observed a large number of proteins covering the fat surface. In particular, it is well known that β-lactoglobulin is able to strongly interact with oil and fat membranes [43,44]. The combination of heat treatment and homogenization may affect the macromolecular arrangement of WP and facilitate the adsorption of protein on the surface of oil droplets by increasing their hydrophobicity [38,45]. Furthermore, in the presence of oils, WPs are able to stabilize the O/W interface by acting as emulsifiers [30,32,37]. An increase in creaming stability, because of the temperature increase during the heat treatment, can also be related to the increased viscosity, as observed in the previous results (Table 3), that improve the physical stability of the system [46].

## 4. Conclusions

The physical properties and stability of WP emulsion gels obtained at increasing temperatures (70, 80, and 90 °C) and pressures (5, 10, and 15 MPa) were investigated. The control sample showed a liquid-like behavior, with the highest and the lowest consistencies and flow behavior indexes compared to the treated samples. On the other hand, in all the treated samples, an elastic behavior was observed, with the elastic modulus (G′) higher than the loss modulus (G″), typical of a viscoelastic gel system. The rheological behavior of the emulsion gels was also strongly dependent on the pressure and temperature conditions applied during the treatment. All the samples showed a bimodal particle size distribution, and by increasing the temperature up to 80 °C, a reduction in *Dv50* values compared to the control samples was observed. All the treated samples showed a higher stability to physical stresses (freezing and centrifugation) than the control samples, with a significant effect of both pressure and temperature on the stability of the samples. The results show the capability to produce gel emulsions with good stability and a broad range of rheological and physical properties by coupling different thermal and mild-pressure treatments. The resulting emulsion gels could be further tested and proposed as innovative food ingredients with good technofunctional properties and be considered to replace fats in a range of foods, such as dairy, bakery, or confectionery products. Further studies would be necessary to validate their effective utilization in food prototypes.

## Figures and Tables

**Figure 1 foods-12-02447-f001:**
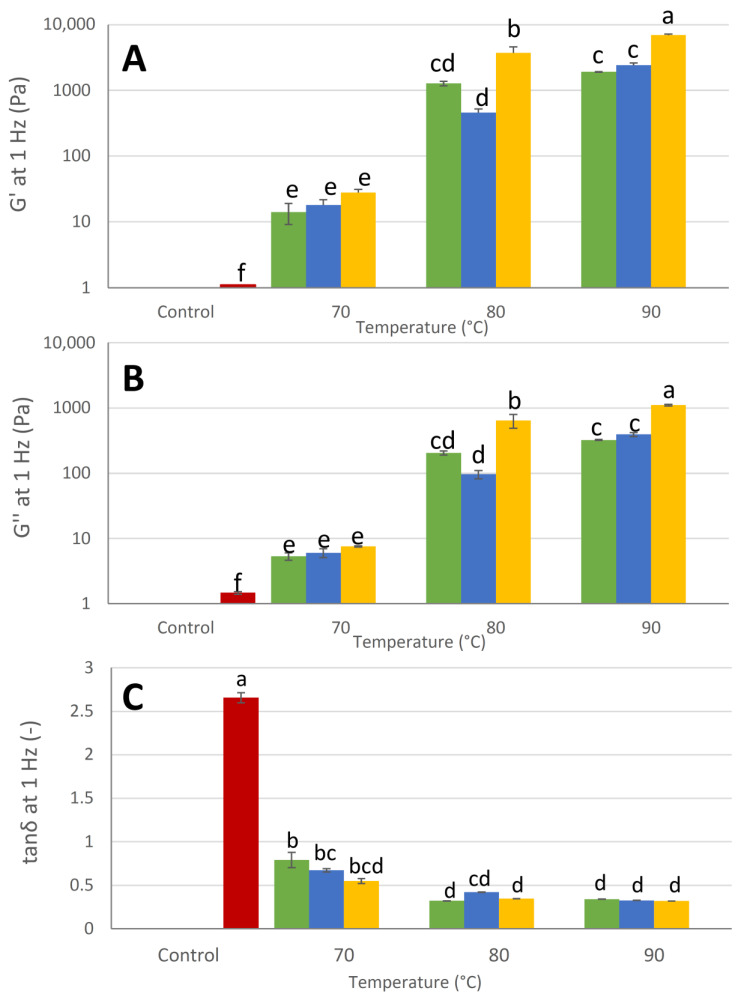
Rheological moduli G′ (**A**), G″ (**B**), and tanδ (**C**) measured at 1 Hz of frequency sweep tests performed on whey protein emulsion gels, produced at different processing temperatures (70, 80, and 90 °C) and pressures (■ 5 MPa, ■ 10 MPa, and ■ 15 MPa). Untreated whey protein gel emulsion was considered as the control of the experiment (■). ^a–f^ Different lowercase letters indicate significant differences (*p* < 0.05) among the samples.

**Figure 2 foods-12-02447-f002:**
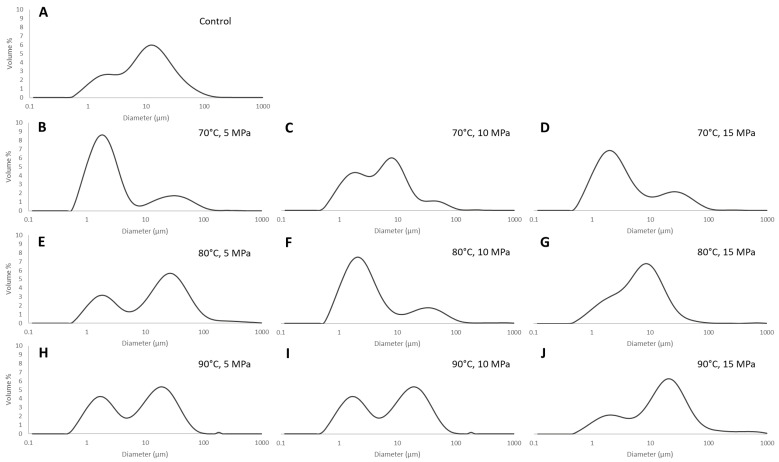
Particle size distribution of whey protein emulsion gels at different processing conditions. (**A**) Control (untreated whey protein emulsion); (**B**) 70 °C, 5 MPa; (**C**) 70 °C, 10 MPa; (**D**) 70 °C, 15 MPa; (**E**) 80 °C, 5 MPa; (**F**) 80 °C, 10 MPa; (**G**) 80 °C, 15 MPa; (**H**) 90 °C, 5 MPa; (**J**) 90 °C, 10 MPa; (**L**) 90 °C, 15 MPa.

**Table 1 foods-12-02447-t001:** Hysteresis area (Pa) and power law parameters (*n, k*) obtained from flow curves of whey protein emulsions. Gelled emulsions were produced at different processing temperatures (70, 80, and 90 °C) and pressures (5, 10, and 15 MPa); an untreated whey protein emulsion was considered as the control of the experiment.

Temperature (°C)	Pressure (MPa)	*n* (-)	k (mPa s)	Hysteresis Area (Pa)
Control		0.737 ± 0.001 ^a^	0.29 ± 0.01 ^e^	0.13 ± <0.01 ^f^
70	5	0.490 ± 0.071 ^b^	1.47 ± 0.01 ^e^	0.34 ± 0.03 ^ef^
	10	0.292 ± 0.012 ^c^	5.48 ± 0.36 ^e^	0.78 ± 0.02 ^ef^
	15	0.305 ± 0.035 ^c^	9.58 ± 0.71 ^e^	0.98 ± 0.11 ^ef^
80	5	0.250 ± 0.014 ^c^	75.69 ± 0.45 ^c^	6.53 ± 0.78 ^d^
	10	0.200 ± 0.014 ^c^	33.68 ± 3.31 ^de^	2.86 ± 0.19 ^e^
	15	0.210 ± 0.028 ^c^	190.08 ± 2.28 ^b^	13.86 ± 0.46 ^c^
90	5	0.216 ± 0.057 ^c^	70.16 ± 3.31 ^cd^	5.50 ± 0.19 ^d^
	10	0.270 ± 0.025 ^c^	102.11 ± 5.97 ^c^	19.92 ± 0.10 ^b^
	15	0.309 ± 0.019 ^c^	355.88 ± 29.39 ^a^	41.03 ± 1.83 ^a^

^a–f^ Different superscripts indicate significant differences (*p* < 0.05) among the samples within the same column.

**Table 2 foods-12-02447-t002:** Particle size parameters (Dv10, Dv50, Dv90, and D_[3,4]_) of WP emulsion gels untreated (control) and treated at different processing temperatures (70, 80, and 90°C) and pressures (5, 10, and 15 MPa).

Temperature (°C)	Pressure (MPa)	Dv10 (μm)	Dv50 (μm)	Dv90 (μm)	D_[3,4]_ (μm)
Control		1.58 ± 0.02 ^a^	10.10 ± 0.07 ^bcd^	41.14 ± 6.54 ^bc^	18.57 ± 4.61 ^bc^
70	5	0.80 ± 0.34 ^b^	2.56 ± 1.92 ^e^	10.96 ± 12.54 ^c^	4.65 ± 4.32 ^c^
	10	0.97 ± 0.11 ^ab^	2.97 ± 0.16 ^e^	36.76 ± 2.76 ^bc^	13.37 ± 0.74 ^bc^
	15	1.11 ± 0.01 ^ab^	8.82 ± 0.5 ^cde^	34.46 ± 6.10 ^bc^	13.99 ± 2.26 ^bc^
80	5	1.46 ± 0.01 ^a^	6.36 ± 0.54 ^cde^	19.34 ± 4.26 ^bc^	10.01 ± 3.02 ^bc^
	10	1.02 ± 0.10 ^ab^	3.76 ± 1.60 ^de^	31.14 ± 3.80 ^bc^	11.44 ± 3.15 ^bc^
	15	0.97 ± 0.04 ^ab^	3.22 ± 0.38 ^e^	33.17 ± 2.92 ^bc^	12.52 ± 0.67 ^bc^
90	5	1.38 ± 0.03 ^ab^	16.13 ± 1.68 ^b^	57.17 ± 13.36 ^ab^	26.01 ± 5.04 ^b^
	10	1.46 ± 0.12 ^a^	11.79 ± 1.18 ^bc^	39.71 ± 17.95 ^bc^	18.85 ± 3.97 ^bc^
	15	1.49 ± 0.09 ^a^	22.66 ± 5.99 ^a^	97.95 ± 32.32 ^a^	43.25 ± 14.50 ^a^

^a–e^ Different superscripts indicate significant differences (*p* < 0.05) among the samples within the same column. D_[3,4]_ represents the volume mean diameter of oil globules. Dv10, Dv50, and Dv90 represent particle sizes in the 10%, 50%, and 90% quantiles of the distribution.

**Table 3 foods-12-02447-t003:** Sedimentation stability (mL of separated water by centrifugation per 100 mL of sample) and emulsion stability (mL of separated oil per 100 mL of sample) of whey protein gel emulsions untreated (control) and obtained at different processing temperatures (70, 80, and 90 °C) and pressures (5, 10, and 15 MPa).

Temperature (°C)	Pressure (MPa)	Sedimentation Stability	Creaming Stability
		(mL of Separated Water/100 mL of Sample)	(mL of Separated Oil/100 mL of Sample)
Control	-	97.85 ± 0.21 ^a^	97.28 ± 0.40 ^a^
70	5	26.78 ± 3.08 ^b^	73.85 ± 0.46 ^b^
	10	15.05 ± 1.31 ^c^	67.74 ± 0.24 ^c^
	15	11.90 ± 0.14 ^c^	59.41 ± 0.83 ^d^
80	5	2.20 ± 0.03 ^d^	27.52 ± 0.70 ^h^
	10	1.66 ± 0.05 ^d^	37.14 ± 0.51 ^g^
	15	0.58 ± 0.11 ^d^	41.07 ± 0.47 ^f^
90	5	1.53 ± 0.03 ^d^	53.52 ± 0.69 ^e^
	10	1.17 ± 0.04 ^d^	41.69 ± 0.03 ^f^
	15	0.13 ± 0.01 ^d^	34.91 ± 1.14 ^g^

^a–h^ Different superscripts indicate significant differences (*p* < 0.05) among the samples within the same column.

## Data Availability

The data presented in this study are available on request from the corresponding author.

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
