# Peer review of "Impact of Combined Thermal Pressure Treatments on Physical Properties and Stability of Whey Protein Gel Emulsions"

_foods, 2023, doi:10.3390/foods12132447_

Round 1

Reviewer 1 Report

The Manuscript is devoted to the investigation of the physical properties and stability of WP emulsion gels obtained using various temperatures and pressures. Overall, the study is novel and well-conceptualized. The results are interpreted well. The study improves general knowledge in this field. However, some minor improvements should be made:

1.      The Authors use the phrase “emulsion droplets” in the Manuscript (lines: 36, 210). I believe that the Authors should be more informative and state that these are oil droplets or droplets of the dispersed phase.

2.      Weigh concentration of a solution is somewhere written as wt/wt, e.g., line 84, and somewhere as w/w, e.g., line 148, in the Manuscript. Please wright the weight concentration of a solution uniformly as w/w throughout the Manuscript.

3.      Line 86: “High-oleic sunflower vegetable oil was obtained from a local market”. Please name the producer, city, and country where the product was manufactured.

4.      Line 94: The following sentence should be clarified: “WPC stock solutions (20% wt/wt) were prepared by powder dissolution in deionized water under mild magnetic stirring for 2 hours at room temperature followed by 24 h at 4°C.” Were the solutions kept refrigerated at 4°C for 24 h after the stirring or were they stirred again, at 4°C for 24 h? Also, please state the stirring speed in rpm.

5.      Line 205: “It may be hypothesized that the application of different homogenization pressures on WP unfolded at different ratios (related to the different heating processes) can have an impact on the protein-protein and protein-oil interface interactions, thus modifying the rheological characteristics of the emulsion gels.” Please clarify this sentence. What do the Authors mean by “the application of different homogenization pressures on WP unfolded at different ratios (related to the different heating processes)”? Was WP unfolded at different ratios based on homogenization pressure and/or different heating processes?

6.      Line 217: “Both treatment temperature and pressure determined a significant effect (P<0.05) on the rheological moduli G’ and G.’’ The Authors probably wanted to say that both temperature and pressure treatment had a significant effect… Please change this.

7.       Line 307: “The formation of intermolecular interactions causes an improvement in the gel network and a greater capacity of the physical structure to retain water molecules.” Does the physical structure have a greater capacity to retain water molecules or oil droplets? Please change this or explain the sentence in the revised Manuscript.

8.      The Authors state in the Manuscript (Abstract, Introduction, Conclusion) that the obtained results are important for the development of novel bakery products. Please explain if is it possible that investigated systems might have even broader applicability in the food industry?

Author Response

We would like to thank the Reviewer for the time and efforts to read and review the paper, and for the constructive comments and revisions; we feel that they significantly helped us in improving manuscript's quality. A detailed, point-by-point response to the Reviewer is reported below (in red).  

The Manuscript is devoted to the investigation of the physical properties and stability of WP emulsion gels obtained using various temperatures and pressures. Overall, the study is novel and well-conceptualized. The results are interpreted well. The study improves general knowledge in this field. However, some minor improvements should be made:

  1. The Authors use the phrase “emulsion droplets” in the Manuscript (lines: 36, 210). I believe that the Authors should be more informative and state that these are oil droplets or droplets of the dispersed phase.

We thank the Reviewer to point this out. We changed the term to "oil droplets"

  1. Weigh concentration of a solution is somewhere written as wt/wt, e.g., line 84, and somewhere as w/w, e.g., line 148, in the Manuscript. Please wright the weight concentration of a solution uniformly as w/w throughout the Manuscript.

We thank the Reviewer to point this out. We uniformed it as recommended to "w/w"

  1. Line 86: “High-oleic sunflower vegetable oil was obtained from a local market”. Please name the producer, city, and country where the product was manufactured.

Thank you for your comment. We added this information to the manuscript.

  1. Line 94: The following sentence should be clarified: “WPC stock solutions (20% wt/wt) were prepared by powder dissolution in deionized water under mild magnetic stirring for 2 hours at room temperature followed by 24 h at 4°C.” Were the solutions kept refrigerated at 4°C for 24 h after the stirring or were they stirred again, at 4°C for 24 h? Also, please state the stirring speed in rpm.

Thank you for your comment. The solutions were kept at 4°C without stirring. The stirring speed was set to around 150 rpm. We modified the sentence in: "WPC stock solutions (20% w/w) were prepared by powder dissolution in deionized water under mild magnetic stirring (150 rpm) for 2 hours at room temperature; subsequently, the stock solution was kept at 4°C for 24 h without stirring."

  1. Line 205: “It may be hypothesized that the application of different homogenization pressures on WP unfolded at different ratios (related to the different heating processes) can have an impact on the protein-protein and protein-oil interface interactions, thus modifying the rheological characteristics of the emulsion gels.” Please clarify this sentence. What do the Authors mean by “the application of different homogenization pressures on WP unfolded at different ratios (related to the different heating processes)”? Was WP unfolded at different ratios based on homogenization pressure and/or different heating processes?

We thank the Reviewer to point this out, as the sentence is not completely clear. We would hypothesize that the application of different heating treatments would possibly result in a different degree of unfolding of the proteins' structure. We would exclude that the homogenization step by itself can cause an (even partial or limited) unfolding phenomenon, because of the relatively low pressure (and energy) involved. In order to better clarify our hypothesis, we modified the sentence in: "It may be hypothesized that the application of different homogenization pressures on previously thermally-unfolded WP ( unfolded at different ratios, in relation to the different heating processes) can have an impact on the protein-protein and protein-oil interface interactions, thus modifying the rheological characteristics of the emulsion gels"

  1. Line 217: “Both treatment temperature and pressure determined a significant effect (P<0.05) on the rheological moduli G’ and G.’’ The Authors probably wanted to say that both temperature and pressure treatment had a significant effect… Please change this.

We thank the Reviewer to point this out. Yes, the Reviewer's interpretation is correct. We thus modified the sentence according to Reviwer's suggestion.

  1.  Line 307: “The formation of intermolecular interactions causes an improvement in the gel network and a greater capacity of the physical structure to retain water molecules.” Does the physical structure have a greater capacity to retain water molecules or oil droplets? Please change this or explain the sentence in the revised Manuscript.

We thank the Reviewer for this comment. We agree that the sentence is not completely clear. We thus modified it in: "The formation of intermolecular interactions causes a rheological improvement in the gel network and a greater capacity of the gel structure to retain water molecules".

  1. The Authors state in the Manuscript (Abstract, Introduction, Conclusion) that the obtained results are important for the development of novel bakery products. Please explain if is it possible that investigated systems might have even broader applicability in the food industry?

We would like to thank the Reviewer for this comment. The developed gel emulsions might have broader applicability in the food industry, for example in the confectionery, or dairy sector (for example in the case of cream cheeses). Of course, the applicability of these gel emulsions should be tested and verified case by case. We reported the bakery sector as an example as we are currently applying these developed ingredients for the production of fat-reduced cookies. However, thanks to this comment, we decided to extend the possible fields of application. 

Reviewer 2 Report

In this research, the authors investigated the influence of homogenization and heat-treatment on the formation and physical stability of WP emulsion gels.

The authors do not specify in section 2.2.2 if the methodology they followed is their own or if they took it from another author. If it was taken from an author, it is important that the source be cited.

It contributes more to the understanding, from the results in section 3.1, of what happens in rheological studies if the authors explain the observed behavior from the chemical point of view.

Why in some emulsions the particle size was bimodal?

It is suggested to the authors to expand the conclusions. Results are observed that lead to good conclusions are not written.

Author Response

We would like to thank the Reviewer for the time and efforts to read and review the paper, and for the constructive comments and revisions; we feel that they significantly helped us in improving manuscript's quality. A detailed, point-by-point response to the Reviewer is reported below (in red).

In this research, the authors investigated the influence of homogenization and heat-treatment on the formation and physical stability of WP emulsion gels.

The authors do not specify in section 2.2.2 if the methodology they followed is their own or if they took it from another author. If it was taken from an author, it is important that the source be cited.

Thank you for your comment. The methodology used to prepare the coarse emulsion, as well as the pilot plant process have been entirely developed by us. 

Further insights about the pilot plant heat exchanger can be viewed in a our previous publication (Rinaldi M.; Cordioli M.; Alinovi M.; Malavasi M.; Barbanti D.; Mucchetti G. Development and Validation of CFD Models of Thermal Treatment on Milk Whey Proteins Dispersion In Batch and Continuous Process Condition. J. Food Eng. 2018, 14, 9–10.). The citation has been reported in the manuscript.

It contributes more to the understanding, from the results in section 3.1, of what happens in rheological studies if the authors explain the observed behavior from the chemical point of view.

Thank you for your suggestion. We tried to improve the section clarity by performing some modification and adding some additional comments to the discussion. 

Why in some emulsions the particle size was bimodal?

Thank you for the comment. It is our opinion that the bimodal distribution is probably related to the observation of gelled emulsion particles having different sizes, i.e. microgels and gelled-particles of bigger dimensions, as similarly stated by other authors in oleogel systems (Muñoz J.; Prieto-Vargas P.; García M.C.; Alfaro-Rodríguez M.C. Effect of a Change in the CaCl2/Pectin Mass Ratio on the Particle Size, Rheology and Physical Stability of Lemon Essential Oil/W Emulgels. Foods 2023, 12, 1137.). We tried to explain better our hypothesis by changing the discussion in:"the presence of a bimodal PSD can be probably caused by the occurrence of an oil emulsion dispersed in a sheared gel, consisting of microgels and gelled-particles of WP having bigger dimensions [35]. "

It is suggested to the authors to expand the conclusions. Results are observed that lead to good conclusions are not written.

Thank you for your comment. Also taking in consideration Reviewer #1 comments, we partiallyv modified the conclusion section